# Human Leukocyte Antigen (HLA) Typing Study Identifies Maternal DQ2 Susceptibility Alleles among Infertile Women: Potential Associations with Autoimmunity and Micronutrients

**DOI:** 10.3390/nu13093270

**Published:** 2021-09-19

**Authors:** Paola Triggianese, Carlo Perricone, Erica De Martino, Arianna D’Antonio, Maria Sole Chimenti, Paola Conigliaro, Sara Ferrigno, Ilio Giambini, Elisabetta Greco, Caterina De Carolis

**Affiliations:** 1Rheumatology, Allergology and Clinical Immunology, Department of “Medicina dei Sistemi”, University of Rome Tor Vergata, 00133 Roma, Italy; paola.triggianese@gmail.com (P.T.); ericadmod@gmail.com (E.D.M.); maria.sole.chimenti@uniroma2.it (M.S.C.); paola.conigliaro@uniroma2.it (P.C.); saraferrigno16@gmail.com (S.F.); elisabetta_greco@yahoo.it (E.G.); cdecarolismd@gmail.com (C.D.C.); 2Rheumatology, Department of Medicine, University of Perugia, Piazzale Giorgio Menghini, 1, 06129 Perugia, Italy; carlo.perricone@gmail.com; 3Department of Laboratory Medicine, Tor Vergata University Hospital of Rome, 00133 Roma, Italy; Ilio.giambini@ptvonline.it

**Keywords:** autoimmunity, infertility, HLA, thyroid, vitamin D

## Abstract

Background. The interplay between female fertility and autoimmune diseases (AIDs) can involve HLA haplotypes and micronutrients. We analyzed the distribution of HLA-DQ2/-DQ8 in women with infertility or recurrent spontaneous abortion (RSA) and possible associations with AIDs and micronutrient status. Methods. Consecutive women (*n* = 187) with infertility and RSA, and controls (*n* = 350) were included. All women were genotyped for HLA-DQ2 (DQA1*0201, A1*05, and B1*02) and -DQ8 (DQA1*03 and DQB1*0302) alleles. Serum 25(OH)D, VB12, folate, and ferritin were evaluated. Results. DQA1*05/B1*02 and the occurrence of at least one DQ2 allele were more prevalent among RSA and infertile women than controls. Infertile women showed lower 25(OH)D and higher prevalence of AIDs than RSA women. In the multivariate analysis, DQA1*05/B1*02 was associated with a significantly higher risk of AIDs in infertile women, and DQA1*05 was independently associated with both 25(OH)D deficiency and AIDs. In RSA women, the presence of AIDs was associated with a significantly higher risk of 25(OH)D deficiency. Conclusion. Our findings showed, for the first time, a higher proportion of DQ2 alleles in infertile and RSA women as compared to controls. Predisposing DQ2 alleles are independent risk factors for AIDs and 25(OH)D deficiency in infertile women and could represent biomarkers for performing early detection of women requiring individually tailored management.

## 1. Introduction

Primary infertility and recurrent spontaneous abortion (RSA) are multifactorial conditions. Approximately 50% of RSA are related to chromosomal abnormalities, such as aneuploidy [1]. However, it is plausible that there are cases of RSA with other genetic association that are not currently routinely detected [1]. Imbalances in the immune system and failure to achieve immune tolerance to the fetus have been implicated as potential causes of “idiopathic” RSA [2,3]. It is known that autoimmune diseases (AIDs) are associated with strong female preponderance and often present during the reproductive years [4]. Several systemic and organ-specific AIDs are associated with infertility and RSA [5,6]. Autoimmune thyroid diseases (AITD), including Hashimoto thyroiditis, can lead to pregnancy-related adverse outcomes due to the direct and indirect effects of autoimmunity and/or mild/subclinical thyroid dysfunction [5,7]. AITD susceptibility is genetically driven [8]. Specific thyroid-associated genes and key genes involved in immunologic pathways are both related with AITD, and Human Leukocyte Antigen (HLA)-DR3 represents the allele carrying the highest risk [8]. As is well-reported, AITD tends to occur with other non-thyroidal AIDs, in particular celiac disease (CD) [9,10,11,12], in which specific HLA haplotypes play a major role.

Micronutrient deficiencies, including 25-hydroxy vitamin D [25(OH)D], vitamin B12 (VB12), folic acid, and iron appear related with poor pregnancy outcomes and are common in AITD and CD [13,14,15,16,17,18,19]. Assessment of the risk factors for adverse maternal and fetal outcomes in women with AIDs is crucial for woman-tailored monitoring plans before and during pregnancy [20]. It is known that micronutrient deficiencies can occur in women of reproductive age, especially during pregnancy, and possible associations between micronutrients and HLA haplotypes are still under investigation [21,22].

Specific HLA-DQ2 alleles (DQA1*0201 and A1*05, B1*02), and HLA-DQ8 alleles (A1*03 and B1*0302) are known to act as high-risk genetic markers for CD, and, as well-documented, symptoms of CD include other AIDs, reproductive failure, and micronutrient deficiencies [13,14,15,16,17,18,19]. Thus, we conducted a cohort study to explore the prevalence of specific HLA-DQ2/-DQ8 alleles in women with primary infertility and RSA without known CD, in order to focus on potential high-risk genetic markers for autoimmunity and micronutrient deficiencies in infertile women. We investigated, for the first time, the prevalence of women carrying “at least one DQ2 allele” by registering women with the occurrence of at least one of the following alleles: DQA1*0201, DQA1*05, and B1*02. In addition, we documented HLA-DQ2 allele positivity by the presence of at least one of the combinations DQA1*0201/B1*02 and DQA1*05/B1*02, and HLA-DQ8 allele positivity by DQA1*03/DQB1*0302. Concomitant AIDs and micronutrient status have been also assessed to explore the interplay between HLA haplotypes, AIDs, and micronutrients in women with reproductive failure.

## 2. Patients and Methods

### 2.1. Study Population

The study included 187 Caucasian women with infertility or RSA, referred to the Polymedical Center for Prevention of Recurrent Spontaneous Abortion, Rome, Italy (2-year recruitment period). The inclusion criteria were: (1) primary infertility or RSA with concomitant AITD and/or anti-phospholipid syndrome (APS) and/or other AIDs and/or with no identifiable causes of infertility/RSA [5,23,24,25]; (2) no history of CD. The control group consisted of 350 women of fertile age recruited at our institution (Tor Vergata University Hospital-PTV, Rome, Italy) with no evidence of serum autoantibodies, micronutrient deficiency, hormonal dysfunctions (prolactin and thyroid disorders), AIDs, and CD.

### 2.2. Clinical Records Collection from the Study Cohort

At enrollment, the clinical records of all the women were registered (Table 1). Infertile women (n = 78) experienced the inability to achieve pregnancy after 1 year of unprotected intercourse, while women with RSA (n = 109) were defined by the occurrence of two or more consecutive failed pregnancies, according to the Practice Committee of the American Society for Reproductive Medicine [5,25].

AIDs included APS, rheumatoid arthritis (RA), Sjögren’s Syndrome (SjS), and undifferentiated connective tissue disease (UCTD), diagnosed in accordance with international criteria [26,27,28,29,30]. Concomitant gastro-intestinal diseases included CD, Crohn’s disease, and autoimmune gastritis, diagnosed in accordance with international criteria [11].

### 2.3. Serum Determinations of Micronutrients

Biochemical parameters were measured on serum from women in the study admitted to our institution (Tor Vergata University Hospital-PTV, Rome, Italy). Serum 25(OH)D, VB12, folate, and ferritin were all measured using the Chemiluminescent Microparticle Immunoassay—CMIA (Architect Instrument, Abbott, Milan, Italy), with the limit of quantitative value of the total serum 25(OH)D, at 2.2 ng/mL at 20% coefficient variation [17].

25(OH)D status was graded as deficiency below 20 ng/mL, while VB12 deficiency was defined below 200 pg/mL. A Folic acid value of ≤ 4 ng/mL was considered deficiency, and serum ferritin levels of ≤ 15 ng/dL were considered iron deficiency [18].

### 2.4. Genotyping of HLA-DQ2 and -DQ8 Alleles

Genomic DNA was extracted from venous EDTA-anticoagulated blood that was drawn and stored at 2–8 °C until genomic DNA extraction. HLA typing was performed on patients’ DNA by using real-time polymerase chain reaction (PCR), according to the manufacturer’s instructions (XeliGen RT System (Eurospital SPA, Italy)) to analyze DQ2 (DQA1*0201, A1*05, and B1*02) and -DQ8 alleles (A1*03 and B1*0302) [31]. HLA-DQ2 allele positivity defined the presence of at least one of the combinations DQA1*0201/B1*02 and DQA1*05/B1*02.

In the study, we also explored the prevalence of women carrying “at least one DQ2 allele” by registering women with the occurrence of at least one of the following alleles: DQA1*0201, DQA1*05, and B1*02.

HLA-DQ8 allele positivity was defined as DQA1*03 with DQB1*0302 [32]. The analyzed alleles were selected based on an extensive review of articles on the association be-tween those risk alleles and AIDs, including AITD, and infertility [8,9,10,11,12,13,14,15,16,17,18,19].

### 2.5. Serology for CD

In HLA-DQ2/-DQ8-positive women, the three serum markers of CD (tissue transglutaminase (TTG) IgA/IgG, endomysial (EMA), and deaminated gliadin peptide (DGP)) were determined to explore undiagnosed CD/celiac autoimmunity [33]. TTG, EMA, and DGP were determined by ELISA kit (Bio-Rad Laboratories, Inc., Segrate, MI, Italia) with normal value absent. The condition of “celiac autoimmunity” was defined as a positive result from serologic tests without endoscopic determination of CD [33].

The study was carried out in accordance with The Code of Ethics of the World Medical Association (Declaration of Helsinki) for experiments involving humans (updated 2008). Informed consent was obtained from patients and controls, and the study was approved by the local ethic committee (No. RS186/16).

### 2.6. Statistical Analysis

Mean and standard deviation (SD) express normally distributed variables. Continuous variables were compared using the parametric unpaired *t* test or the nonparametric Mann–Whitney U test when appropriate. Categorical variables were presented with absolute frequencies and percentages and were compared using the Chi-squared test or Fisher’s exact test when appropriate. Multivariate analyses were used to evaluate the association between variables in infertile and RSA women. In the current study, the independent variables we used were: HLA-DQ2 (DQA1*0201, A1*05, and B1*02), and HLA-DQ8 (A1*03 and B1*0302) susceptibility alleles (present vs. absent), micronutrient deficiency (25(OH)D (deficiency vs. normal range) and VB12 (deficiency vs. normal range)), and autoimmune diseases (AITD (present vs. absent) and AIDs (present vs. absent)).

*p* values < 0.05 were considered significant. All statistical analyses were performed using GraphPad Prism version 8.2 (GraphPad Software, San Diego, CA, USA).

## 3. Results

### 3.1. Patient Population

RSA and infertile women had a similar mean age compared with controls (Table 1). RSA women experienced mainly 2 SA (*n* = 70, 64.2%), while RSA women with >2 SA were 35.8% (*n* = 39).

Infertile women showed a higher prevalence of AIDs and AITD than RSA (*p* = 0.03 (OR 1.92, 95% C.I. 1–3.56), *p* = 0.01 (OR 2.2, 95% C.I. 1.15–4.12), respectively (Table 1)). Systemic AIDs were similarly distributed among infertile and RSA women (Table 1). Autoimmune gastritis occurred in two infertile and three RSA women; of these five patients, three (two infertile, one RSA) also had a concomitant AITD.

### 3.2. Micronutrients Status

The levels of 25(OH)D in the infertile group were significantly lower than in the RSA group and in controls (*p* = 0.04 for both the comparisons); the proportion of 25(OH)D deficiency in women with infertility (35/78, 44.87%) was higher than that of the RSA group (33/109, 30.28%; *p* = 0.04, OR 1.87, 95% C.I. 1–3.34). Four RSA women showed VB12 deficiency. Only one infertile woman and four RSA women had a folate deficiency. Iron deficiency was similarly observed in infertile and RSA women (26 vs. 34%). Among women with iron deficiency, a concomitant AITD was more frequently observed in infertile than in RSA women (53.8% vs. 22.2%, *p* = 0.045).

No differences in the prevalence of deficiencies of 25(OH)D, VB12, iron, and folate occurred in RSA women in accordance with the number of SA (women with 2 SA versus women with >2 SA, data not shown).

### 3.3. HLA-DQ2/-DQ8 Distribution

HLA-DQA1*0201/B1*02 showed a similar prevalence between controls (78/350, 22.3%), RSA (15/109 = 13.7%), and infertile women (11/78 = 14.1%). HLA-DQA1*05/B1*02 haplotype had a higher prevalence in RSA (26/109, 23.85%) and infertile women (16/78, 20.5%) compared with controls (37/350, 10.57%; *p* = 0.001, OR 2.6, 95% C.I. 1.5–4.6; *p* = 0.02, OR 2.2, 95% C.I. 1.14–4.16, respectively) (Figure 1). HLA-DQ8 allele positivity showed a similar prevalence in all the women in the study (controls, 48/350, 13.7%; RSA 13/109, 11.9%; infertile 9/78, 11.5%). No significant differences in the prevalence of DQ2 and DQ8 positivity resulted when comparing infertile with RSA women. No differences in the distribution of HLA-DQ2/-DQ8 occurred in RSA women in accordance with the number of SA (data not shown).

We also analyzed the prevalence of women carrying at least one of the following DQ2 alleles: DQA1*0201, DQA1*05, and B1*02. A significantly higher prevalence of at least one DQ2 allele was observed in RSA (75/109, 67.9%) and infertile women (60/78, 69.23%) than in controls (101/350, 28.8%; *p* < 0.0001, OR 5.4, 95% C.I. 3.4–8.5; *p* < 0.0001, OR 8.2, 95% C.I. 4.6–14.9, respectively) (Figure 1).

### 3.4. HLA-DQ2/-DQ8 Distribution and AIDs

The occurrence of at least one of the alleles DQA1*0201, DQA1*05, and B1*02 has been registered at a higher prevalence in women with AIDs than in women without AIDs among infertile women (*p* = 0.04, OR 3.2, 95% C.I. 1–8.8). Moreover, RSA women with AITD showed at least one of the alleles DQA1*0201, DQA1*05, and B1*02 at a higher prevalence than women with other AIDs (*p* = 0.01, OR 7, 95% C.I. 1.5–25). Moreover, a higher proportion of women with AIDs and at least one concomitant DQA1*0201, DQA1*05, and B1*02 has been documented in infertile women compared with RSA (*p* = 0.04, OR 0.3, 95% C.I. 0.1–0.9). Among infertile women, HLA-DQA1*05/B1*02 was more prevalent in women with AIDs than in those without (*p* = 0.009, OR 4.6, 95% C.I. 1.4–14) as well as in women with AITD than in those with other AIDs (*p* = 0.04, OR 7.3, 95% C.I. 1.12–84). All the subgroups among evaluated women exhibited a similar prevalence of HLA-DQ8 alleles (Table 2).

### 3.5. HLA-DQ2/-DQ8 Positivity and CD

The frequencies of TTG, EMA, and AGA were determined in HLA-DQ2/-DQ8-positive women. Approximately 3% of RSA (4/109, 3.7%) and infertile women (3/78, 3.8%) had seropositivity without endoscopic confirmation (“celiac autoimmunity”). All seropositive patients were also HLA-DQ2/-DQ8-positive. In the study, none of the seven women with celiac autoimmunity underwent endoscopic determination of CD. A new diagnosis of a defined CD was performed in one infertile woman with clinical symptoms, CD seropositivity, and endoscopic confirmation of CD.

In the multivariate analysis, HLA-DQ2 positivity (defined as DQA1*05/B1*02) was associated with a significantly higher risk of AIDs (*p* = 0.02, OR 24, 95% C.I. 0.17–340) in women with infertility. Moreover, in infertile women, DQA1*05 was independently associated with 25(OH)D deficiency (*p* = 0.04, OR 5.4, 95% C.I. 1–37) and AIDs (*p* = 0.045, OR 5.5, 95% C.I. 1–37.3). In the RSA group, the presence of AIDs was associated with a significantly higher risk of 25(OH)D deficiency (*p* = 0.015, OR 5, 95% C.I. 1–30).

## 4. Discussion

In this study, a higher proportion of HLA-DQ2 serotype was observed in women with infertility or RSA than in controls in the absence of confirmed CD. For the first time, we showed that HLA-DQ2/-DQ8 alleles have a similar prevalence in women with primary infertility compared to women with RSA. Infertile women also have a greater prevalence of AITD.

HLA expression in tissues at the maternal–fetal interface plays a key role for the success of pregnancy and the development of the fetus [34]. However, the link between specific-susceptibility HLA alleles and the pathogenesis of pregnancy failure is still debated [35].

Taken together, our findings are in accordance with data reporting a significantly increased prevalence of HLA-DQ2-positivity in women with infertility and RSA compared to healthy controls, supporting a possible pathogenic link between these susceptibility HLA alleles and the early stages of pregnancy [36]. As is known, several autoimmune disorders tend to occur together, and the close relationships between specific autoimmune diseases can be largely explained by a shared genetic background. In this context, the HLA antigens DQ2 (DQA1*0201, A1*05, B1*02) and DQ8 (A1*03 and B1*0302) are the major common genetic predisposition. These HLA alleles carrying the high risk for AIDs have been associated not only with CD but also with AITD and other non-thyroidal AIDs [9,10,11,12]. Moreover, CD, AITD, and AIDs can be associated with obstetrical complications by several mechanisms [5,7,11,12,19,37,38,39,40]. Thus, the rationale for detecting those specific genotypes goes along the direction of exploring the distribution of potential genetic risk markers for autoimmunity and micronutrient deficiencies in women with infertility and RSA. In particular, we also investigated, for the first time, the prevalence of women carrying at least one of the DQ2 alleles DQA1*0201, DQA1*05, and B1*02 in order to explore the prevalence of these selected DQ2 HLA haplotypes in infertile women, in accordance with the idea of a shared genetic background among AIDs, micronutrient status, and women’s fertility.

Case–control and cohort studies showed that women with infertility and RSA have a nearly six-fold increased risk of CD [41]. Several reports documented reproductive dysfunction, such as delayed menarche, RSA, and infertility in untreated women with CD [42]. The mechanisms by which CD can cause obstetrical complications include the occurrence of micronutrient deficit, including 25(OH)D, VB12, folate, and ferritin [39,40]. Specifically, a 25(OH)D deficiency status has been previously described in women with primary infertility [18,43]. Accordingly, our data confirm these findings, providing evidence of significantly lower 25(OH)D levels in infertile women than in RSA women and controls, while VB12, folate, and ferritin had overall normal concentrations in all studied women. Other mechanisms by which CD can influence fertility and pregnancy outcome include TTG inhibition of syncytial-TTG, resulting in impairment of placental development [44,45]. In our study population, an already documented CD at enrollment represented an exclusion criterion for the study because of the well-known association between high-risk HLA DQ2/DQ8 alleles and CD. Nevertheless, in the study, we explored the occurrence of the serum markers of CD in HLA DQ2/DQ8-positive women with infertility/RSA. As documented by authors in the literature, the condition of positive results from serologic tests for CD without endoscopic determination of CD is called “celiac autoimmunity” and, to date, the clinical outcomes of celiac autoimmunity have not been thoroughly evaluated. Choung et al. showed that subsets of adults with seropositive results do not progress toward symptomatic CD but spontaneously become negative for CD despite consuming gluten-containing foods [33]. However, we investigated the proportion of women with celiac autoimmunity in our cohort of infertile/RSA women in order to describe the potential occurrence of such condition in women with reproductive failure. We registered approximately 3% of RSA and infertile women with celiac autoimmunity. At the same time, a new diagnosis of a defined CD was performed in one infertile woman with clinical symptoms, CD seropositivity, and endoscopic confirmation of CD. The overall prevalence (7/187) of celiac autoimmunity in infertile/RSA women in our study results is similar to that from cohorts of patients with AIDs and higher than that found in the general population. At the time of the study, none of the seven women with celiac autoimmunity underwent endoscopic determination of CD. As mentioned, it is disputed whether CD seropositivity is predictive of clinically evident CD and its possible comorbidities, thus a tailored prospective long-term follow-up of these women will be mandatory in our clinical practice [46,47,48,49].

Almost half of the infertile/RSA cohort showed at least one autoimmune disease, mainly including AITD. In this study, AITD has been analyzed separately from other AIDs because of the evidence from the literature supporting the role of AITD in pregnancy outcome and in both fetal and maternal health [50,51]. As is known, thyroid autoantibodies exert their effect in a TSH-dependent and TSH-independent manner [24,52,53,54]. RSA associated with anti-thyroid antibody positivity may be due to the direct effect of anti-thyroid antibodies on the fetal-placental tissue. Moreover, anti-thyroid antibodies interact with placental hormones, especially with chorionic gonadotropin and with chorionic thyrotropin, leading to an alteration of their function [52,54]. Thyroid disorders, both autoimmunity and/or dysfunction, may also be considered as a detectable marker of maternal immunological alterations, given the associations between thyroid homeostasis and immune abnormalities [52,53,54,55,56]. Micronutrients, including 25(OH)D, VB12, folic acid, and iron are often found to be deficient in AITD, resulting in malfunctioning of the thyroid. Interestingly, in our group of infertile women with iron deficiency, a concomitant AITD occurred at a higher prevalence than in RSA women. As is known, HLA class II allele distribution confers susceptibility to AITD in terms of both mono- and polyglandular autoimmunity [57]. Moreover, the risk of recurrence of subacute thyroiditis has been reported to be HLA-dependent [58]. Accordingly, a higher prevalence of HLA-DQ2 positivity occurred in infertile women with AITD more frequently compared to women with non-thyroidal-AIDs. As is known, HLA molecules play a crucial role in the development of AIDs. AIDs, in our study cohort, included APS, RA, SjS, and UCTD; moreover, concomitant Crohn’s disease and autoimmune gastritis were registered. These concomitant AIDs, which rarely occurred among the women in the study, are all known to be associated with DQ2 susceptibility alleles [59]. The presence of anti-phospholipid and/or anti-nuclear antibodies has been described as more frequent in infertile women and is probably responsible for fertility disorders [60,61]. However, pregnancy outcome in women with RA, SjS, UCTD, and other rheumatic diseases is still under investigation [32,62,63]. APS is certainly the AID mostly associated with obstetrical complications [64]. Interestingly, authors reported that untreated CD patients may show an increased prevalence of anti-cardiolipin antibodies [65]. The comparison of HLA alleles in infertile/RSA women with or without AIDs has been performed to explore a possible different distribution of specific HLA risk alleles and concomitant AIDs in accordance with infertility/RSA status. A higher proportion of women with at least one DQ2 allele and concomitant AIDs was identified in infertile women compared with RSA women, suggesting the hypothesis that primary infertility is associated with predisposing HLA-DQ2 alleles that expose women to elevated risk of AIDs. Along this direction, among infertile women, at least one DQ2 allele was identified in a higher prevalence in women with concomitant AIDs than in women without AIDs. Nevertheless, in women with RSA, at least one DQ2 allele was more prevalent in women with AITD than in those with other AIDs, supporting the role of susceptibility HLA alleles for AITD and the early stages of pregnancy in RSA [36].

In addition, we, interestingly, for the first time documented that in women with primary infertility, DQ2 alleles are independently associated with 25(OH)D deficiency and AIDs, while in RSA women, 25(OH)D deficiency was associated with the presence of AIDs, suggesting the potential role of other risk alleles in this context. In addition, our findings described a lower mean 25(OH)D level in infertile women than in both RSA and fertile controls, according to data from the literature [18]. Concerns remain regarding circulating 25(OH)D levels measured, in the present study, by CMIA that, as described by authors, overestimates insufficient values [66]. The gold standard method for 25(OH)D measurement is the high-pressure liquid chromatography-tandem mass spectrometry, but the CMIA is the method used in our hospital laboratory in accordance with the balancing of many factors, including availability and feasibility to use [66]. So, clinicians must be prudent in the assessment of 25(OH)D levels, as variations exist between the assay methods [66].

## 5. Conclusions

Taken together, our findings support the hypothesis that primary infertility is associated with predisposing HLA-DQ2 alleles that expose women to an elevated risk of micronutrient deficiency and AIDs, including AITD. In this view, women’s infertility shows typical autoimmune footprints involving selected DQ2 genetic determinants of 25(OH)D deficiency and thyroid dysfunctions, thus taking the fashion of a defined autoimmune disease. In our study cohort, women with CD were not investigated. However, further studies concerning potential links between micronutrients, AIDs, and HLA in large cohorts of CD patients are awaited, to explore how our findings might be applied in CD women.

We suggest that micronutrient status and presence of concomitant AIDs should be deeply investigated in women of reproductive age, and possibly supplementation of 25(OH)D could be advised. In addition, HLA genotyping for identification of predisposing DQ2 susceptibility alleles could be suggested in selected cases. DQ2 alleles could represent potential biomarkers to perform an early detection of women requiring individually tailored management. Further investigations involving larger cohorts are awaited to enrich the diagnostic panel for women with reproductive failure.

## Figures and Tables

**Figure 1 nutrients-13-03270-f001:**
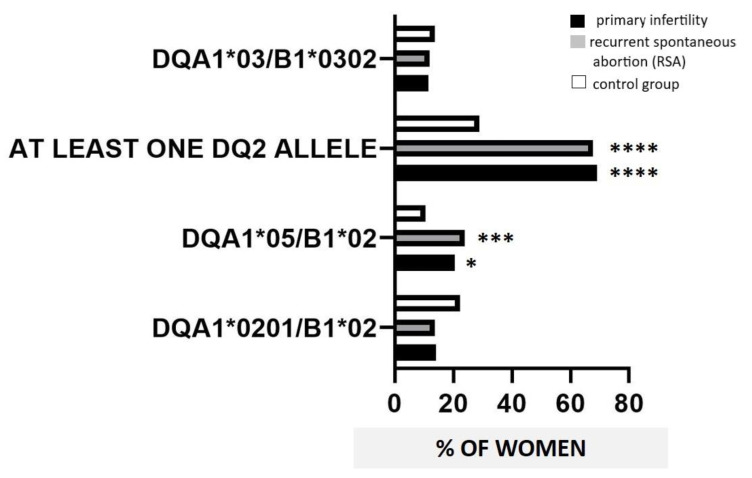
HLA distribution in the study population. Black bars, primary infertility; grey bars, recurrent spontaneous abortion (RSA); white bars, control group. DQA1*03 with DQB1*0302 defined “HLA-DQ8 alleles positivity”. The notation “at least one DQ2 allele” described the occurrence of at least one of the following alleles: DQA1*0201, DQA1*05, and B1*02. HLA-DQ2 allele positivity defined the presence of at least one of the combinations: DQA1*0201/B1*02 and DQA1*05/B1*02. A significantly higher prevalence of at least one DQ2 allele was revealed in infertile and RSA women than in controls. HLA-DQ2 positivity as DQA1*05/B1*02 showed a higher prevalence in infertile and RSA women compared with controls. Chi-squared test or Fisher’ exact test were used to compare groups (* *p* < 0.05; *** *p* ≤ 0.001, **** *p* ≤ 0.0001 compared with controls).

**Table 1 nutrients-13-03270-t001:** Data from women with infertility and recurrent spontaneous abortion.

	Primary Infertility (*n* = 78)	RSA (*n* = 109)	Controls (*n* = 350)	*p* Value
Age (mean ± SD)	37.3 ± 4.5	37 ± 4.3	37 ± 5	Ns
N of SA (mean ± SD)	N.A.	2.6 ± 0.6	N.A.	N.A.
GW of SA (mean ± SD)	N.A.	8 ± 2	N.A.	N.A.
25(OH)D (mean ± SD, ng/mL) ^§^	21.8 ± 10 *	25.3 ± 11	26 ± 5.6	0.04
VB12 (mean ± SD, pg/mL) ^§^	450 ± 181	400 ± 132	460 ± 120	Ns
Folate (mean ± SD, ng/mL) ^§^	17.2 ± 16	18.9 ± 22	19 ± 9	Ns
Ferritin (mean ± SD, ng/dl) ^§^	30 ± 21.6	29.3 ± 21.7	34 ± 22.8	Ns
AIDs (N/%)	38/48.7 **	36/33	N.A.	0.03
Other AIDs (N/%)	13/16.7	15/13.7	N.A.	Ns
AITD (N/%)	29/37.2 **	23/21.1	N.A.	0.01
APS (N/%)	5/6.4	10/9.2	N.A.	Ns
RA (N/%)	3/3.8	1/0.9	N.A.	Ns
SjS (N/%)	1/1.3	2/1.8	N.A.	Ns
UCTD (N/%)	2/2.6	2/1.8	N.A.	Ns
G.I. diseases (N/%)	4/5.13	3/2.7	N.A.	Ns

RSA, recurrent spontaneous abortion; SA, spontaneous abortion; GW, gestation week; AIDs, autoimmune diseases; AITD, autoimmune thyroid disease; APS, antiphospholipid syndrome; RA, rheumatoid arthritis; SjS, Sjögren’s syndrome; UCTD, undifferentiated connective tissue disease; G.I., gastro-intestinal diseases including Celiac Disease (*n* = 1, infertile), Crohn’s disease (*n* = 1, infertile), and autoimmune gastritis (*n* = 2 infertile, *n* = 3 RSA). ^§^ Healthy population norms: 25(OH)D = 30–100 ng/mL, VB12 = 187–883 pg/mL, folate = 3.5–20.5 ng/mL, ferritin = 21.8–274.6 ng/mL (measured using the Chemiluminescent Microparticle Immunoassay, CMIA). N.A., not applicable; Ns, not significant. Continuous variables were expressed as mean and standard deviation (SD) and compared by *t* test or Mann–Whitney U test, when appropriate. Categorical variables were presented with absolute frequencies and percentages and compared using the Chi-squared test or Fisher’ exact test when appropriate (*p* values < 0.05 between the groups; * compared with both RSA and controls; ** compared with RSA).

**Table 2 nutrients-13-03270-t002:** HLA distribution in patients with or without autoimmune diseases.

		DQA1*0201/B1*02	DQA1*05/B1*02	HLA-DQ8	At Least One DQ2
AIDs	RSA (*n* = 36)	6/16.7	10/27.8	10/27.8	24/66.7
	Infertile (*n* = 38)	6/15.8	12/31.6 **	7/18.4	33/86.8 *
No AIDs	RSA (*n* = 73)	9/12.3	16/21.9	16/21.9	51/69.9
	Infertile (*n* = 40)	5/12.5	4/10 **	9/22.5	27/67.5 *
AITD	RSA (*n* = 23)	5/21.7	9/39.1	5/21.7	19/82.6 **
	Infertile (*n* = 29)	6/20.7	11/37.9 *	5/17.2	26/89.6
Other	RSA (*n* = 15)	1/6.7	2/13.4	5/33.4	6/40 **
	Infertile (*n* = 13)	1/7.7	1/7.7 *	2/13.4	10/76.9

HLA-DQ2 allele positivity defined the presence of at least one of the combinations: DQA1*0201/B1*02 and DQA1*05/B1*02, while HLA-DQ8 allele positivity was defined as DQA1*03/DQB1*0302. The notation “at least one DQ2 allele” described the occurrence of at least one of the following alleles: DQA1*0201, DQA1*05, and B1*02. RSA, recurrent spontaneous abortion; AIDs, autoimmune diseases; AITD, autoimmune thyroid disorders; Other, non-thyroidal AIDs. Groups were compared with Fisher’s exact test (AIDs vs. No AIDs; AITD vs. Other; * *p* < 0.05; ** *p* ≤ 0.01).

## Data Availability

The data presented in this study are available on request from the authors.

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
