# Peer review of "Human Leukocyte Antigen (HLA) Typing Study Identifies Maternal DQ2 Susceptibility Alleles among Infertile Women: Potential Associations with Autoimmunity and Micronutrients"

_nutrients, 2021, doi:10.3390/nu13093270_

Round 1

Reviewer 1 Report

Title: Human leukocyte antigen (HLA) typing study identified maternal DQ2 susceptibility allels among infertile women: potential associations with autoimmunity and micronutrients.

Summary: The authors performed genotyping of HLA-DQ2 (DQA1*0201/B1*02 and A1*05/B1*02) and HLA-DQ8 (DQA1*03/DQB1*0302) in different female populations (healthy control, autoimmune diseases, autoimmune thyroid diseases, recurrent spontaneous abortion, and primary infertility). The results were analyzed to derive association between the genotype and disease.

Comments:
1. The rationale for detecting those specific genotypes (e.g. HLA-DQ2..) is lacking. How many types of alleles in HLA, in addition to HLA-DQ2, in the study group? Please provide evidence supporting the link between those alleles and diseases (genome wide association study or HLA analysis).

2. Why autoimmune thyroid diseases (AITD) is separated from autoimmune diseases (AIDs) in this study? What are the other types of autoimmune diseases in the AIDs?

3. It is not clear what is the purpose to comparison between RSA and infertile group in each autoimmune in table 2.

4. The authors are encouraged to describe clearly the rationale for focusing on specific genotypes.

Author Response

Point-by-point reply to Reviewers

Reviewer 1.

R1. The rationale for detecting those specific genotypes (e.g. HLA-DQ2..) is lacking. How many types of alleles in HLA, in addition to HLA-DQ2, in the study group? Please provide evidence supporting the link between those alleles and diseases (genome wide association study or HLA analysis).

Authors: In accordance with this comment, we have clarified the rationale in both introduction and discussion of the revised paper by including evidence supporting the idea of a link between selected alleles and the diseases.

R2. Why autoimmune thyroid diseases (AITD) is separated from autoimmune diseases (AIDs) in this study? What are the other types of autoimmune diseases in the AIDs?

Authors: In the revised version of the paper, we accordingly added comments on the separated analysis of AITD with the respect to the other AIDs and we also included details on the other types of AIDs occurring in the cohort study.

R3. It is not clear what is the purpose to comparison between RSA and infertile group In each autoimmune in table 2.

Authors: The purpose to compare RSA with infertile women has been stated in the revised version of the discussion.

R4. The authors are encouraged to describe clearly the rationale for focusing on specific genotypes.

Authors: In accordance with this comment, we better described in the revised paper the rationale for focusing on specific genotypes by providing evidence from the literature supporting the link between infertility, AIDs (including AITD), micronutrients, and HLA genotypes.

Reviewer 2 Report

The manuscript by Triggianese et al. entitled “ Human Leukocyte Antigen (HLA) typing study identifies maternal DQ2 susceptibility alleles among infertile women: potential associations with autoimmunity and micronutrients” describes an interesting issue of HLA association with micronutrients deficiency and autoimmunity in infertile women. The general impression of the manuscript is good, but still, there is some important issue that should be addressed by the authors.

First of all, I had some difficulties in following the patient's section, as some of the inclusion criteria are a bit unintelligible. The inclusion criteria presented in the first paragraph of the Patients and Methods are as follows:

1) a condition of primary infertility or RSA with no identifiable causes but AITD or anti-phospholipid syndrome (APS) [21, 22];

2) no history of CD.

Based on this presentation I understood that only patients with the diagnosed autoimmune thyroid diseases and the antiphospholipid syndrome were included in the study.

The confusion arises in the third paragraph of the same section where authors list the autoimmune diseases… Then when the reader moves to the results section get more confused as the results are discussed according to the presence of AIDs (table 1 and 2 also presents results relying on the occurrence of AIDs in the examined cohorts of patients).

So, is the presence of any AID exclusion criteria or not? It should be explained!

Maybe the Authors diagnosed patients enrolled in the study to check how many patients are suffering from AID? If so, please describe the diagnosing process.

Other concerns in the Patients and Method section:

  • Where are the subsections of this section? The text should be divided according to the experimental procedures
  • “Concomitant gastro-intestinal diseases included CD, Crohn’s dis-ease, and autoimmune gastritis [11].” – I have an impression that there is something missing
  • Please provide details of experimental procedures: high-resolution genotyping of HLA alleles, serum markers evaluation, estimation of micronutrients levels in patients
  • Insert norms of micronutrients levels in Table 1.

Result Section:

  • Please provide data for the control group in Table 1. Were the levels of micronutrients evaluated for the control group? I know that these women were healthy with no micronutrients deficiency, but such data would show the reader what is the distribution of such values in the healthy cohort.
  • “HLA-DQA1*0201/B1*02 showed a similar prevalence between controls (78/350, 22.3%), RSA (15/109 = 13.7%) and infertile women (11/78 = 14.1%). HLA-DQA1*05/B1*02 haplotype had a higher prevalence in RSA (26/109, 23.85%) and infertile women (16/78, 20.5%) compared with controls (37/350, 10.57%; P=0.001, OR 2.6, 95% C.I. 1.5-4.6; P=0.02, OR 2.2, 95%C.I. 1.14-4.16, respectively).” – where are these data presented any graph or table?
  • “HLA-DQ8 alleles positivity showed a similar prevalence in all the women in the study (controls, 48/350, 13.7%; RSA 13/109, 11.9%; infertile 9/78, 11.5%). No significant differences in the prevalence of DQ2 and DQ8 positivity resulted when comparing infer-tile with RSA women.” – this sentence would better fit as a continuation of the first section of the “HLA-DQ2/-DQ8 distribution”
  • “ No difference in the distribution of HLA-DQ8 positivity was observed between women with AITD and women with other AIDs (Table 2).” – I do not understand this sentence
  • “In the multivariate analysis, HLA-DQ2 positivity (defined as DQA1*05/B1*02) was associated with a significantly higher risk of AIDs (P=0.02, OR 24, 95% CI 0.17-340) in women with infertility. Moreover, in infertile women, DQA1*05 was independently as-sociated with 25(OH)D deficiency (P=0.04, OR 5.4, 95% CI 1-37) and AIDs (P=0.045, OR 5.5, 95% CI 1-37.3). In the RSA group, the presence of AIDs was associated with a signifi-cantly higher risk of 25(OH)D deficiency (P=0.015, OR 5, 95% CI 1-30).” – how this applies to CD?
  • Where is the proper presentation of micronutrients analysis?

Other issues that should be addressed:

  • Why did the Authors decide to analyse only HLA-DQ2/-DQ8 alleles? It should be explained in the introduction.
  • Have the authors compared their data with the number of RSA?
  • What was the reason for primary infertility in women enrolled in the study? Can the Authors include the source of primary infertility in the analysis?
  • “ As mentioned, we found 7/187 patients with CD seropositivity that is dis-puted whether is predictive of clinically evident CD” – I do not understand this sentence.
  • “Either thyroid specific or genes in immunologic pathways are related with AITD,”- I do not understand this sentence

Small language issues to correct:

Abstract – please correct the last part of the following sentence: “identify early women more likely to need tailored management”

Introduction – I would advise using another world instead of aggregate in the following sentence:  “As well reported, AITD tend to aggregate with other non-thyroidal AIDs, in particular celiac disease (CD) [9-12] in which specific HLA haplotypes play a major role.”

Figure 1. Please insert the legend with bar colours, it would facilities following the presented results.

Author Response

R1. First of all, I had some difficulties in following the patient's section, as some of the inclusion criteria are a bit unintelligible. The inclusion criteria presented in the first paragraph of the Patients and Methods are as follows: 1) a condition of primary infertility or RSA with no identifiable causes but AITD or anti-phospholipid syndrome (APS) [21, 22]; 2) no history of CD. Based on this presentation I understood that only patients with the diagnosed autoimmune thyroid diseases and the antiphospholipid syndrome were included in the study. The confusion arises in the third paragraph of the same section where authors list the autoimmune diseases… Then when the reader moves to the results section get more confused as the results are discussed according to the presence of AIDs (table 1 and 2 also presents results relying on the occurrence of AIDs in the examined cohorts of patients).

So, is the presence of any AID exclusion criteria or not? It should be explained!

Maybe the Authors diagnosed patients enrolled in the study to check how many patients are suffering from AID? If so, please describe the diagnosing process.

Authors: In accordance with these comments, we deeply revised the text in order to clarify both inclusion criteria and methods. As stated, women with RSA/infertility have been included in the study in both the occurrence of AIDs (including APS) or in the case of an unexplained reproductive failure (no identifiable causes). So, the presenze of an AID was not an exclusion criterion. Moreover, women have not been diagnosed to have a concomitant AID/AITD during the study but these data have been registered in the clinical records collection step. So, our study methods did not include the diagnosing process for AIDs/AITD.

R2. Where are the subsections of this section? The text should be divided according to the experimental procedures

Authors. In accordance with the Reviewer 2, we added in the revised paper specific subsections in the “Patients and Methods” section.

R2. “Concomitant gastro-intestinal diseases included CD, Crohn’s dis-ease, and autoimmune gastritis [11].” – I have an impression that there is something missing

Authors: We accordingly quoted adequated references.

R2. Please provide details of experimental procedures: high-resolution genotyping of HLA alleles, serum markers evaluation, estimation of micronutrients levels in patients

Authors: In accordance with the Reviewer 2, in the revised paper we provided details on experimental procedures.

R2. Insert norms of micronutrients levels in Table 1.

Authors: In the revised version, the Table 1 has been improved according to the Reviewer 2.

R2. Result Section:

R2. Please provide data for the control group in Table 1. Were the levels of micronutrients evaluated for the control group? I know that these women were healthy with no micronutrients deficiency, but such data would show the reader what is the distribution of such values in the healthy cohort.

Authors: In the revised version, the Table 1 has been improved according to the suggestions of the Reviewer 2 by adding data from controls.

R2. “HLA-DQA1*0201/B1*02 showed a similar prevalence between controls (78/350, 22.3%), RSA (15/109 = 13.7%) and infertile women (11/78 = 14.1%). HLA-DQA1*05/B1*02 haplotype had a higher prevalence in RSA (26/109, 23.85%) and infertile women (16/78, 20.5%) compared with controls (37/350, 10.57%; P=0.001, OR 2.6, 95% C.I. 1.5-4.6; P=0.02, OR 2.2, 95%C.I. 1.14-4.16, respectively).” – where are these data presented any graph or table?

Authors: In the revised version, the text has been changed by adding "Figure 1".

R2. “HLA-DQ8 alleles positivity showed a similar prevalence in all the women in the study (controls, 48/350, 13.7%; RSA 13/109, 11.9%; infertile 9/78, 11.5%). No significant differences in the prevalence of DQ2 and DQ8 positivity resulted when comparing infer-tile with RSA women.” – this sentence would better fit as a continuation of the first section of the “HLA-DQ2/-DQ8 distribution”

Authors: In the revised version, this sentence has been added as a continuation of the first section of the “HLA-DQ2/-DQ8 distribution

R2. “ No difference in the distribution of HLA-DQ8 positivity was observed between women with AITD and women with other AIDs (Table 2).” – I do not understand this sentence

Authors: In the revised version, this sentence has been changed ("All the subgroups among evaluated women exhibited a similar prevalence of HLA-DQ8 alleles (Table 2)").

R2. “In the multivariate analysis, HLA-DQ2 positivity (defined as DQA1*05/B1*02) was associated with a significantly higher risk of AIDs (P=0.02, OR 24, 95% CI 0.17-340) in women with infertility. Moreover, in infertile women, DQA1*05 was independently as-sociated with 25(OH)D deficiency (P=0.04, OR 5.4, 95% CI 1-37) and AIDs (P=0.045, OR 5.5, 95% CI 1-37.3). In the RSA group, the presence of AIDs was associated with a signifi-cantly higher risk of 25(OH)D deficiency (P=0.015, OR 5, 95% CI 1-30).” – how this applies to CD?

Authors: In our study cohort, women with CD have not been investigated. However, further investigations from large cohort studies on links between MNDs, AIDs and HLA in CD patients are required. In the revised version, this sentence has been added in the concluding remarks of the discussion.

R2. Where is the proper presentation of micronutrients analysis?

Authors: Micronutrients have been analyzed (as stated in the "Statistical Analysis") as mean and standard deviation (if normally distributed) and compared among groups using the parametric unpaired T test or the nonparametric Mann–Whitney U test (when appropriate). The prevalence of MNDs were presented with absolute frequencies and percentages and were compared between groups using the Chi-squared test or Fisher’s exact test when appropriate. Multivariate analyses were used to evaluate the association between variables in infertile and RSA women. In the current study, the independent variables we used were: HLA-DQ2 (DQA1*0201, A1*05, B1*02), and HLA-DQ8 (A1*03, B1*0302) susceptibility alleles (present vs absent), micronutrients deficiency [25(OH)D (deficiency vs normal range), VB12 (deficiency vs normal range)], and autoimmune diseases [AITD (present vs absent), and AIDs (present vs absent)]. Laboratory measurements of micronutrients have been better described in the revised version of the paper.

R2. Why did the Authors decide to analyse only HLA-DQ2/-DQ8 alleles? It should be explained in the introduction.

Authors: In the revised version, we added comments on the rationale underlying such a choice.

R2. Have the authors compared their data with the number of RSA?

Authors: In the revised version, we added specific sentences in the "Results" concerning data (not shown) on prevalence of MNDs in RSA women in accordance with the number of SA as well as on the HLA distribution.

R2. What was the reason for primary infertility in women enrolled in the study? Can the Authors include the source of primary infertility in the analysis?

Authors: According to the comments, we better clarified that infertile women were enrolled in the study in order to explore the idea of an autoimmune/genetic background of primary (unexplained) infertility as stated in introduction and discussion. As women with primary infertility have a condion with no identifiable causes, the specific source of primary infertility cannot be added in the analysis.

R2. “ As mentioned, we found 7/187 patients with CD seropositivity that is dis-puted whether is predictive of clinically evident CD” – I do not understand this sentence.

Authors: In the revised version, this sentence has been modified (In our cohort, 7/187 women revealed a CD seropositivity: as already mentioned, it is disputed whether the CD seropositivity is predictive of clinically evident CD [46-48]. Choung et al. showed that a subset of adults with seropositive results does not progress toward symptomatic CD but spontaneously becomes negative for CD despite consuming gluten-containing foods [33]. In this view, a tailored prospective long-term follow-up of these women will be mandatory in our clinical practice.).

R2. “Either thyroid specific or genes in immunologic pathways are related with AITD,”- I do not understand this sentence

Authors: In the revised version, this sentence has been modified.

R2. Small language issues to correct:

Abstract – please correct the last part of the following sentence: “identify early women more likely to need tailored management”; Introduction – I would advise using another world instead of aggregate in the following sentence:  “As well reported, AITD tend to aggregate with other non-thyroidal AIDs, in particular celiac disease (CD) [9-12] in which specific HLA haplotypes play a major role.”

Authors: In the revised version, these small language issues have been corrected.

R2. Figure 1. Please insert the legend with bar colours, it would facilities following the presented results.

Authors: In the revised version, we inserted the legend with bar colours in order to facilitate the lecture of the presented results.

Round 2

Reviewer 1 Report

  1. It is difficult to know how the author address the previous issues without referring the page number and line number. Authors are encouraged to include the information in the Point-by-point reply.
  2. Based on the description in the introduction, the reason why the authors selected the 2 HLA-DQ2 alleles is association to a non-thyroidal AIDs, celiac disease (CD). But in the manuscript, it is no clear to me what is the percentage of the CD in each group (Primary infertility, RSA, controls) in this study. The authors did state that 7/187 women revealed a CD seropositivity (page 6, line 237). It is still difficult to understand the rationale and conclusion from this study.
  3. What does the “at least one DQ2” mean in the Figure 1 and Table 2? Based on the description of the manuscript. DQ2 consists of DQA1*0201/B1*02 and DQA1*05/B1*02. Therefore, the idea of “at least one DQ2” should be the summation of the two population, isn’t it? Why the case number in the “At least one DQ2” is larger than the combination of the two individual groups? Again, it is difficult to understand the rationale and conclusion from this study.

Author Response

R1. It is difficult to know how the author address the previous issues without referring the page number and line number. Authors are encouraged to include the information in the Point-by-point reply.

Authors: Dear Reviewer 1, all the changes in the revised manuscript have been coloured in yellow colour in order to highlight all the revisions. However, in accordance with your last suggestion, we included in the point-by-point reply, pages and line numbers of all the changes made.

Page 1, lines 29-30; page 2, lines 45-47; page 2, lines 57-70; page 2, lines 75-77; page 3, lines 95-97; page 3, lines 101-102; page 3, lines 106-115; page 4, lines 134-139; page 5, lines 181-183; page 5, lines 189-196; page 6, lines 203-205; page 6, lines 211-218; page 6, lines 221-222; page 6, lines 224-227; page 6, lines 235-236; page 7, lines 258-272; page 7, lines 284-297; page 8, lines 298-310; page 8, lines 327-330; page 8, lines 337-348; page 9, lines 352-361; page 9, lines 367-370; page 9, lines 375-376; page 10, lines 423-425;  page 11, lines 463-464; page 12, lines 547-549.

R2. Based on the description in the introduction, the reason why the authors selected the 2 HLA-DQ2 alleles is association to a non-thyroidal AIDs, celiac disease (CD). But in the manuscript, it is no clear to me what is the percentage of the CD in each group (Primary infertility, RSA, controls) in this study. The authors did state that 7/187 women revealed a CD seropositivity (page 6, line 237). It is still difficult to understand the rationale and conclusion from this study.

Authors: An already documented CD at the enrollement represented an exclusion criterion for the study, as stated in the “Methods”, because of the well-known association between high-risk HLA DQ2/DQ8 alleles and CD. Nevertheless, we explored the occurrence of the serum markers of CD (TTG, EMA, and DGP) in the included women. As documented from authors in the literature, a condition of positive results from serologic tests for CD without endoscopic determination of CD is called “celiac autoimmunity” and, to date, clinical outcomes of celiac autoimmunity have not been thoroughly evaluated. However, we investigated the proportion of women with celiac autoimmunity in our cohort of infertile/RSA women in order to describe the potential occurrence of such condition in women with reproductive failure. We registered approximately 3% of RSA (4/109, 3.7%) and infertile women (3/78, 3.8%) with celiac autoimmunity. At the same time, a new diagnosis of a defined CD was performed in 1 infertile woman with clinical symptoms, CD seropositivity, and endoscopic confirmation for CD. The overall prevalence (7/187) of infertile/RSA women with celiac autoimmunity is similar to that from cohorts of patients with AIDs and higher than that found in the general population. At the study, none of 7 women with celiac autoimmunity underwent endoscopic determination of CD.

In the revised version of the paper, we made changes in order to better describe that part of the study (pages 7-8, lines 284-306).

R3. What does the “at least one DQ2” mean in the Figure 1 and Table 2? Based on the description of the manuscript. DQ2 consists of DQA1*0201/B1*02 and DQA1*05/B1*02. Therefore, the idea of “at least one DQ2” should be the summation of the two population, isn’t it? Why the case number in the “At least one DQ2” is larger than the combination of the two individual groups? Again, it is difficult to understand the rationale and conclusion from this study.

Authors: We are grateful to the Reviewer 1 for all the suggestions. In the revised manuscript we better described the notation of “at least one DQ2” and the rationale of its analysis in the study. We adequately made changes in the introduction, methods, results (including Figure 1 and Table 2, as suggested) and discussion (all the revisions have been coloured in yellow).

In particular, in the introduction (page 2, lines 57-70) we changed the text as follows: “Specific HLA-DQ2 alleles (DQA1*0201, A1*05, B1*02), and HLA-DQ8 alleles (A1*03, B1*0302) are known to act as high-risk genetic markers for CD, and, as well documented, symptoms of CD include other AIDs, reproductive failure, and micronutrients deficiencies [13-19]. Thus, we conducted a cohort study to explore the prevalence of specific HLA-DQ2/-DQ8 alleles in women with primary infertility and RSA without known CD in order to focus on potential high-risk genetic markers for autoimmunity and micronutrients deficiencies in infertile women. We investigated for the first time the prevalence of women carrying “at least one DQ2 allele” by registering women with the occurrence of at least one of the following alleles: DQA1*0201, DQA1*05, B1*02. In addition, we documented HLA-DQ2 alleles positivity by the presence of at least one of the combinations DQA1*0201/B1*02, DQA1*05/B1*02, and HLA-DQ8 alleles positivity by DQA1*03/ DQB1*0302. Concomitant AIDs and micronutrients status have been also assessed to explore the interplay between HLA haplotypes, AIDs, and micronutrients in women with reproductive failure.” In the methods session on “Genotyping of HLA-DQ2 and –DQ8 alleles”, we added: “In the study, we also explored the prevalence of women carrying “at least one DQ2 allele” by registering women with the occurrence of at least one of the following alleles: DQA1*0201, DQA1*05, B1*02.” (page 4, lines 127-129). In the results (page 5, lines 195-196), we added: “We also analyzed the prevalence of women carrying at least one of the following DQ2 alleles: DQA1*0201, DQA1*05, B1*02.”; we also made changes at lines 1-8 of the paragraph “HLA-DQ2/-DQ8 distribution and AIDs” (page 6, lines 211-218). In the discussion (lines 258-272, page 7), we changed the text as follows: “As known, several autoimmune disorders tend to occur together and the close relation-ships between specific autoimmune diseases can be largely explained by a shared genetic background. In this context, the HLA antigens DQ2 (DQA1*0201, A1*05, B1*02), and DQ8 (A1*03, B1*0302) are the major common genetic predisposition. These HLA alleles carry-ing the high risk for AIDs have been associated not only with CD but also with AITD and other non-thyroidal AIDs [9-12]. Moreover, CD, AITD and AIDs can be associated with ob-stetrical complications by several mechanisms [5, 7, 11, 12, 19, 37-40]. Thus, the rationale for detecting those specific genotypes goes along the direction of exploring the distribution of potential risk genetic markers for autoimmunity and micronutrients deficiencies in women with infertility and RSA. In particular, we also investigated for the first time the prevalence of women carrying at least one of the DQ2 alleles DQA1*0201, DQA1*05, B1*02 in order to explore the prevalence of these selected DQ2 HLA haplotypes in infertile wom-en in accordance with the idea of a shared genetic background among AIDs, micronutri-ents status and women fertility.”

In addition, several changes have been adequately made at pages 7-9, lines 284-305, 308-310327-331, 337-348, 351-361, 367-370, 375-376.

In order to reply to the suggestions from the Reviewer 1 (“Therefore, the idea of “at least one DQ2” should be the summation of the two population, isn’t it? Why the case number in the “At least one DQ2” is larger than the combination of the two individual groups?”), in the revised version we better stated definitions and notations - in particular concerning “at least one DQ2 allele” -  in order to highlight that, among all women in the study, there are subjects who had only one of the selected DQ2 allele or more than one of them (DQA1*0201, A1*05, B1*02), together with subjects with at least one of the combinations DQA1*0201/B1*02, DQA1*05/B1*02. So, the subgroup “at least one DQ2 allele” included all the individual cases.

According to the Reviewer 1 who considered difficult to understand the rationale and conclusion from our study based on the previous version of the paper, we provided more details, in the revised version, on the rationale of the study concerning the interplay between HLA haplotypes (DQ2 susceptibility alleles), AIDs, and micronutrients in infertile/RSA women, by improving the text in the introduction, methods, results, discussion, legends of Figure 1 and Table 2.

As stated in conclusion, taken together, our findings support the hypothesis that primary infertility is associated with predisposing HLA-DQ2 alleles that expose women at elevated risk of autoimmune diseases and micronutrients deficiency and. In this view, women infertility shows typical autoimmune footprints involving selected DQ2 genetic determinants thus taking the fashion of a defined autoimmune disease.

Reviewer 2 Report

I would like to thank the Authors for their effort put into the improvement of their manuscript. Still, there are few things that should be considered before acceptance.

First of all, I do really want the Authors to include details regarding methods, especially NGS and micronutrients determination.  And provide these details in separate sections for micronutrients analysis and NGS.

I have an impression that the Authors omitted the introduction of micronutrients’ norms in table 1.

Please provide the results regarding micronutrients in a separate subsection of Results.

Please correct the last part of the following sentence from the Abstract: “identify early women more likely to need tailored management”.

Author Response

I would like to thank the Authors for their effort put into the improvement of their manuscript. Still, there are few things that should be considered before acceptance.

R1. First of all, I do really want the Authors to include details regarding methods, especially NGS and micronutrients determination. And provide these details in separate sections for micronutrients analysis and NGS.

Authors: In accordance with these comments, we deeply revised the text of methods in order to include details regarding micronutrients, serology of CD, and genetic analysis. We also provided these details in separate sections for each determination.

In the revised paper, we made the following changes (pages 3-4, lines 110-139):

Serum determinations of micronutrients

Biochemical parameters were measured on serum from women in the study admit-ted to our Institution (Tor Vergata University Hospital-PTV, Rome, Italy). Serum 25(OH)D, VB12, folate, and ferritin were all measured using the Chemiluminescent Microparticle Immunoassay - CMIA (Architect Instrument, Abbott, Milan, Italy), with the limit of quanti-tative value of the total serum 25(OH)D, at 2.2 ng/mL at 20% coefficient variation [17].

25(OH)D status was graded as deficiency below 20 ng/ml while VB12 deficiency was defined below 200 pg/mL. Folic acid value ≤ 4 ng/mL was considered as deficiency, serum ferritin levels ≤ 15 ng/dL were considered iron deficiency [18].

Genotyping of HLA-DQ2 and –DQ8 alleles

Genomic DNA was extracted from venous EDTA-anticoagulated blood that was drawn and stored at 2–8 °C until genomic DNA extraction. HLA typing was performed on patients DNA by using real-time polymerase chain reaction (PCR) according to the manu-facturer’s instructions [XeliGen RT System (Eurospital)] to analyze DQ2 (DQA1*0201, A1*05, B1*02), and -DQ8 alleles (A1*03 and B1*0302) [31]. HLA-DQ2 alleles positivity de-fined the presence of at least one of the combina-tions: DQA1*0201/B1*02, DQA1*05/B1*02.

In the study, we also explored the prevalence of women carrying “at least one DQ2 al-lele” by registering women with the occurrence of at least one of the following alleles: DQA1*0201, DQA1*05, B1*02.

HLA-DQ8 alleles positivity was defined as DQA1*03 with DQB1*0302 [32]. The ana-lyzed alleles were selected based on an extensive review of articles on the association be-tween those risk alleles and AIDs, including AITD, and infertility [8-19].

Serology for CD

In HLA-DQ2/-DQ8 positive women, the three serum markers of CD [tissue transglu-taminase (TTG) IgA/IgG, endomysial (EMA), and deaminated gliadin peptide (DGP)] were determined to explore undiagnosed CD/celiac autoimmunity [33]. TTG, EMA, and DGP were determined by ELISA kit (Bio-Rad Laboratories, Inc., Segrate, MI, Italia) with normal value absent. A condition of “celiac autoimmunity” was defined as positive results from serologic tests without endoscopic determination of CD [33].

R2. I have an impression that the Authors omitted the introduction of micronutrients’ norms in table 1.

Authors: In accordance with this comment, we added the healthy population norms of micronutrients in the legend of table 1 (25(OH)D = 30-100 ng/ml, VB12 = 187- 883 pg/ml, folate = 3.5-20.5 ng/ml, ferritin = 21.8-274.6 ng/ml). Moreover, we provided additional comments on analytical techniques for micronutrients (mainly vitamin D) in discussion section (“In addition, our findings described a lower mean 25(OH)D level in infertile women than in both RSA and fertile controls according to data from the literature [18]. Concerns remain regarding circulating 25(OH)D levels measured, in the present study, by CMIA that, as described by authors, overestimates insufficient values [66]. The gold standard method for 25(OH)D measurement is the high-pressure liquid chromatography-tandem mass spectrometry but the CMIA is the methods used in our hospital laboratory in accordance with the balancing out many factors including availability and feasibility to use [66]. So, clinicians must be prudent in the assessment of 25(OH)D levels as variations exist between the assay methods [66]”. – page 9, lines 352-361)

R3. Please provide the results regarding micronutrients in a separate subsection of Results.

Authors: In accordance with the Reviewer, in the revised version of the paper we provided the results regarding micronutrients in a separate subsection of Results.

R4. Please correct the last part of the following sentence from the Abstract: “identify early women more likely to need tailored management”.

Authors: In accordance with this comment, we changed the selected sentence from the abstract.

(“perform an early detection of women requiring an individually tailored management”)
